Mating system of Datura inoxia: association between selfing rates and herkogamy within populations

http://orcid.org/0000-0003-2050-9026 Jiménez-Lobato Vania 1 2
Núñez-Farfán Juan 2 farfan@unam.mx
1 Escuela Superior de Desarrollo Sustentable, Universidad Autónoma de Guerrero , Cátedras CONACYT , México
2 Laboratorio de Genética Ecológica y Evolución, Departamento de Ecología Evolutiva, Instituto de Ecología, Universidad Nacional Autónoma de México , Ciudad de México, Distrito Federal , Mexico
Vision Todd
Electronic publication date: 2021 Mar 19
Publication date: 2021
Volume: 9
Electronic Location ID: e10698
Received 2020 Sep 9; Accepted 2020 Dec 13
Copyright: © 2021 Jiménez-Lobato and Núñez-Farfán
Copyright year: 2021
Copyright holder: Jiménez-Lobato and Núñez-Farfán
License: This is an open access article distributed under the terms of the Creative Commons Attribution License, which permits unrestricted use, distribution, reproduction and adaptation in any medium and for any purpose provided that it is properly attributed. For attribution, the original author(s), title, publication source (PeerJ) and either DOI or URL of the article must be cited.
License URL: https://creativecommons.org/licenses/by/4.0/

Keywords: Chihuahuan desert, Datura inoxia, Genetic variance, Herkogamy, Inbreeding coefficient, Inbreeding depression, Plant mating system, Outcrossing rate, Primary selfing rate, Selfing syndrome

Funding: Consejo Nacional de Cienica y Tecnología of México (CONACyT) 0154P-N9506 CONACyT 81490 Cátedra CONACyT ID7902 This work was supported by the Consejo Nacional de Cienica y Tecnología of México (CONACyT), grant 0154P-N9506 to Juan Núñez-Farfán. The CONACyT scholarship No. 81490 and the Cátedra CONACyT ID7902 granted to Vania Jiménez-Lobato. The funders had no role in study design, data collection and analysis, decision to publish, or preparation of the manuscript.

==============================
Plant mating system determines, to a great extent, the demographic and genetic properties of populations, hence their potential for adaptive evolution. Variation in plant mating system has been documented between phylogenetically related species as well between populations of a species. A common evolutionary transition, from outcrossing to selfing, is likely to occur under environmental spatial variation in the service of pollinators. Here, we studied two phenotypically (in floral traits) and genetically (in neutral molecular markers) differentiated populations of the annual, insect-pollinated, plant Datura inoxia in Mexico, that differ in the service of pollinators (Mapimí and Cañada Moreno). First, we determined the populations’ parameters of phenotypic in herkogamy, outcrossing and selfing rates with microsatellite loci, and assessed between generation (adults and seedlings) inbreeding, and inbreeding depression. Second, we compared the relationships between parameters in each population. Results point strong differences between populations: plants in Mapimí have, on average, approach herkogamy, higher outcrossing rate (tm = 0.68), lower primary selfing rate (r = 0.35), and lower inbreeding at equilibrium (Fe = 0.24) and higher inbreeding depression (δ = 0.25), than the populations of Cañada. Outcrossing seems to be favored in Mapimí while selfing in Cañada. The relationship between r and Fe were negatively associated with herkogamy in Mapimí; here, progenies derived from plants with no herkogamy or reverse herkogamy had higher selfing rate and inbreeding coefficient than plants with approach herkogamy. The difference Fe–F is positively related to primary selfing rate (r) only in Cañada Moreno which suggests inbreeding depression in selfing individuals and then genetic purging. In conclusion, mating system evolution may occur differentially among maternal lineages within populations of Datura inoxia, in which approach herkogamy favors higher outcrossing rates and low levels of inbreeding and inbreeding depression, while no herkogamy or reverse herkogamy lead to the evolution of the “selfing syndrome” following the purge of deleterious alleles despite high inbreeding among individuals.

Introduction

Plant mating systems affect the demographic dynamics and genetic structure of plant populations, and thus their adaptive potential (Charlesworth & Charlesworth, 1979; Eckert et al., 2009). Mating systems often vary widely among populations of self-compatible species, from predominant or complete selfing to complete outcrossing. Intermediate (or mixed) mating systems, that is, those that combine selfing and outcrossing strategies, are also common in nature (Stebbins, 1957; Goodwillie, Kalisz & Eckert, 2005; Barrett, 2010).

The transition from outcrossing to selfing is one of the most frequent evolutionary shifts in flowering plants (Stebbins, 1957; Barrett, 2010; Sicard & Lenhard, 2011). Evidence of this phenomenon comes from phylogenetic studies (Goodwillie, 1999; Foxe et al., 2009), investigating differences within and among species, and at different geographic scales (Duncan & Rausher, 2013; Wright, Kalisz & Slotte, 2013).

Two main scenarios are thought to favor the evolution of selfing. One is the transmission advantage of selfers relative to outcrossers (3:2 gametes), which would promote the spread of selfing modifiers in a population unless prevented by natural selection (Fisher, 1941). The second scenario considers that under ecological conditions that are unfavorable for cross-pollination (i.e., rarity or absence of potential mates and/or pollen vectors), natural selection would favor reproductive assurance through selfing (Baker, 1955; Stebbins, 1957; Schoen & Lloyd, 1992; Schoen, Morgan & Bataillon, 1996). Despite the potential transmission and reproductive advantages, selfing as a mating system may also restrict gene flow within and among populations. Consequently, selfing might increase levels of inbreeding and homozygosity, and ultimately the likelihood of inbreeding depression (δ) in populations (Charlesworth & Charlesworth, 1987; Charlesworth & Wright, 2001; Takebayashi & Morrell, 2001). Therefore, the inbreeding coefficient (F) is expected to correlate with historical level of selfing in a population. In the long term, however, theoretical models and empirical evidence suggest that increasing levels of inbreeding promote selection against inbred individuals by purging populations of lethal or deleterious alleles, thus reducing genetic load (Husband & Schemske, 1996; Takebayashi & Delph, 2000; Crnokrak & Barrett, 2002; Morran, Parmenter & Phillips, 2009). Hence, populations with a long history of inbreeding are expected to have lower levels of inbreeding depression (Yahara, 1992)

Herkogamy, the spatial segregation of sex organs within the flower, is considered a main floral trait affecting selfing rates (Webb & Lloyd, 1986). In hermaphroditic flowers, the likelihood of selfing is reduced when the stigma surpasses the length of the anthers (i.e., approach herkogamy), because flower stigmas are prevented from receiving self-pollen. In presence of pollinators, this also provides the opportunity for outcross pollen to first reach the stigmas. In contrast, when the anthers and stigmas are placed at the same level (i.e., no—herkogamy) or the stigmas are below the anthers (i.e.,, reverse herkogamy), selfing is likely to occur, with (facilitated) or without (autonomous) pollinators (Lloyd, 1992). At the population level, the relationship between herkogamy and selfing/outcrossing rates may be an indicator of the historical reliability of pollination (Opedal, 2018). Within-population, this relationship may also be explained by other causal mechanisms. For instance, stigma clogging may increase competition between self- and cross-pollen in non-herkogamous flowers, favoring higher outcrossing rates in herkogamous flowers, or just because self-pollen deposition during pollinator visits creates a “shield” against interference from heterospecific pollen (Opedal, 2018). Also, a positive correlation between approach herkogamy and floral traits related to pollinator attraction, contributes to the maintenance of the relationship between herkogamous flowers and an outcrossing strategy (Opedal, 2018). In other species herkogamy may reduce the interference between sexual functions enhancing pollen export efficiency (Lloyd & Webb, 1986; Barrett, 2002). Further, reverse or no herkogamy might increase reproductive fitness (i.e., seed number) and thus be favored by selection, but it should also positively correlate with selfing rate. Contrary, natural selection might favor/maintain approach herkogamy thus increasing outcrossing rate (Motten & Stone, 2000; Elle & Hare, 2002; Herlihy & Eckert, 2007).

Herkogamy is a complex trait determined by genetic and ecological factors (Ashman & Majetic, 2006; Opedal et al., 2017). Within populations, standing genetic variation in herkogamy would depend on the history of selection of herkogamy on individual lineages, and its adaptive value could depend upon selection pressures exerted by pollinators—particularly when the abundance of pollinators fluctuates in time—or to other selective factors. When associations among selfing rate (s), inbreeding coefficient (F) and herkogamy are maintained over time, within-population covariance among these variables may be established at the lineage level. Significant associations between high outcrossing rate and approach herkogamy have been reported before (Takebayashi, Wolf & Delph, 2006; De Vos et al., 2018; but see Chen et al., 2009; Brys & Jacquemyn, 2012; Opedal, Armbruster & Pélabon, 2015; Toräng et al., 2017). However, few studies have evaluated the association between herkogamy and outcrossing rate within-populations (Epperson & Clegg, 1987; Motten & Antonovics, 1992; Carr & Fenster, 1994; Karron et al., 1997; Brunet & Eckert, 1998; Takebayashi & Delph, 2000; Elle & Hare, 2002; Medrano, Herrera & Barret, 2005; Herlihy & Eckert, 2007) and even fewer have considered relationships among herkogamy, inbreeding coefficient and mating system.

Many annual, self-fertilizing, plant species evolve in stressful environments (e.g., disturbed and/or arid habitats) where pollinator abundance is reduced and/or unpredictable (Friedman & Rubin, 2015). Under variable pollination service, daily and seasonal fluctuations present different opportunities for cross-or self-fertilization maintaining variation on traits that affect the plants’ mating system, such as herkogamy or the inbreeding history of each maternal lineage (Schoen & Lloyd, 1992; Schoen, Morgan & Bataillon, 1996; Morgan & Wilson, 2005; Eckert, Samis & Dart, 2006; Barrett, 2010; Shirk & Hamrick, 2014; Pannell, 2015). However, it is not clear how strong and how frequently the association between mating strategies, herkogamy and inbreeding history of lineages occur within populations in short-lived species. Here, we assessed the within-population mating system dynamics, with replication in two populations of the annual plant Datura inoxia Mill. We focus on the within-population level and less on among-population expectations in order to detect if individuals with more herkogamous flowers show a history of less inbreeding associated to lower levels of selfing. Until now, relative few studies (see review in Opedal, 2018) have assessed these associations within (rather than among) populations.

Materials and Methods

Study species and sampled populations

Datura inoxia (Solanaceae) is a summer annual, self-compatible plant that inhabits arid and semi-arid lands in Mexico and Southern USA (i.e., The Chihuahuan desert) (Barclay, 1959; Lockwood, 1973). High daily fluctuation of ambient temperature is a characteristic of these ecosystems. The onset of flowering of D. inoxia is in July and lasts through September, similar to other Datura species (Bronstein et al., 2009). Datura inoxia produces large, funnel-shaped, and nectar producing hermaphroditic flowers that open at dusk and remain receptive for only one night. During the flowering period, individuals can display from a few up to tens of flowers each night. Flower traits related to mating system such as herkogamy and flower size, vary widely within and among populations (Jiménez-Lobato & Núñez-Farfán, 2012). Further, comparisons between genetic differentiation at neutral loci (FST) and phenotypic differentiation (QST) in floral traits suggest adaptive evolution under diverging selection (Jiménez-Lobato & Núñez-Farfán, 2012). Flowers are commonly visited by honeybees, which collect pollen, and hawkmoths (Manduca sexta, M. quinquemaculata and Hyles lineata) which forage for nectar (Barclay, 1959; Lockwood, 1973; Grant, 1983; McCall et al., 2018; V. Jiménez-Lobato, personal observation).

Two populations of D. inoxia were selected to assess the relationships between herkogamy, selfing rate, and inbreeding. A previous report (Jiménez-Lobato & Núñez-Farfán, 2012) indicated substantial within-population variation in herkogamy and flower size. The Cañada de Moreno population (CM) is located in the State of Querétaro (21° 17′ 43″ N; 100° 31′ 00″ W) in the Mexican Bajío at 1,933 m a.s.l. During the flowering period of D. inoxia (July–September), this locality has an average temperature of 18.8 °C, with a daily range from 7.4 °C to 30.8 °C, and a three-month total precipitation of 314 mm. The Mapimí population (Map) is located at 1,157 m a.s.l. in the Mexican Plateau in the States of Coahuila and Durango (26° 41′ 11″ N; 103° 44′ 49″ W). This is a more xeric environment, with a quarterly average temperature of 22.8 °C (range: 12.6–32.9 °C) and a total three-month precipitation of 253 mm (Meteorological Service of Mexico: http://smn.cna.gob.mx/). Collection of seed material for experimental analyses was made under the permission SGPA-DGGFS-712-1596-17 (Subsecretaría de Gestión para la Protección Ambiental, Secretaría de Medio ambiente y Recursos Naturales, Mexico).

Variation in the level of herkogamy within and among populations

Thirty reproductive plants were randomly selected and tagged for sampling in a 1 ha area within each population. For each individual plant, 4–6 open flowers were randomly selected to measure herkogamy. Herkogamy was calculated as the difference between pistil and stamen length. Approach herkogamy was defined as the pistil surpassing the stamens in length (henceforth “positive herkogamy”), whereas reverse herkogamy was the opposite trend (henceforth “negative herkogamy”). Absence of herkogamy occurred when pistil and stamens had equal lengths.

Mating system parameters

To estimate mating system parameters from each marked plant in the field, five mature fruits derived from natural pollination were collected, labelled and bagged. In the laboratory, seeds of each fruit were separated and germinated in a greenhouse, and seeds of each fruit within each family (maternal plant) were sown in separate pots. Germination per fruit per family was recorded for 30 days. To obtain an average estimate of germination rate per fruit, we recorded the final number of seeds germinated in each pot. Germination percentage was ≥90% (se = 3.18) for all plants. Once seedlings emerged, we collected leaf tissue from young plants, bagged, labelled, and stored in an ultra-freezer at −97 °C. Finally, we analyzed mating system parameters in 20 seedlings from each of 30 maternal families per population (N = 600).

DNA was extracted from seedlings following the Miniprep protocol (Doyle & Doyle, 1987). Five microsatellite nuclear loci developed for D. stramonium (Andraca, 2009) were amplified for each seedling. Further, we standardized one additional microsatellite locus for D. inoxia (F8: Rw: 5′ -GGACAACATCTTTGCGACCC- 3′), in order to obtain a total of six polymorphic microsatellite loci per individual plant. Primers were labelled with PET, VIC, 6-FAM, and NED dyes (Applied Biosystems, Foster City, CA, USA) (see Supplemental Information: PCR protocols).

Multilocus outcrossing (tm) and selfing (s = 1 − tm) rates, primary selfing rate (r) and inbreeding coefficient (F) were estimated for each maternal family for each population. Mating system parameters (tm and s) at the family level were estimated with MLTR 3.2 (Ritland, 2002) using the Expectation-Maximization method, which allows missing data and undetected null alleles (Ritland & Jain, 1981). Standard errors and standard deviations were estimated by bootstrapping, with 1,000 replicates and re-sampling individuals at the family level. The frequency of null alleles per locus, per population, was assessed using Micro-Checker v.2.2.3 (Cock et al., 2004). One locus (G8) did not amplify for any of the plants from CM, so analyses were carried out with five loci in that population, and six loci in Map. Selfing rates (s) obtained from molecular markers, after fertilization and germination, might not be completely independent from inbreeding depression and thus may underestimate its true value (Lande, Schemske & Schultz, 1994). Primary selfing rate (r) is a better predictor of mating system because it assesses separately the magnitude of inbreeding depression. Hence, r refers to the proportion of selfed progeny at the time of fertilization (Lande, Schemske & Schultz, 1994). The primary selfing rate (r) was calculated for each maternal family as: r = s/[1 − δ + sδ], where s is the selfing rate obtained from microsatellite loci, and δ is the cumulative inbreeding depression obtained for each population (see below).

The inbreeding coefficient (F) may include components of inbreeding other than just self- or cross- fertilization, such as biparental inbreeding or population substructure. Here, we used F as a proxy of the inbreeding history of each maternal family (i.e., adult cohort) (Latta & Ritland, 1994). Inbreeding coefficient (F) values were inferred for each lineage from the microsatellite loci amplified previously, with GenePop v.4.2 (Rousset & Raymond, 1995; Rousset, 2008). To investigate whether inbreeding depression was associated with the selfing rate and herkogamy, we calculated inbreeding coefficients at equilibrium (Fe) (i.e., progeny cohort) for each lineage, assuming that adult F and tm are constant among generations (Ritland, 1990). We then related these differences with selfing rate and herkogamy. Since Fe increases in relation to F due to self-fertilization, larger differences between F and Fe at each maternal lineage should indicate the presence of higher inbreeding depression. Once selection against inbred progeny has occurred, Fe and F will be equal (Ritland, 1990; Shirk & Hamrick, 2014). This approach yields an estimation of potentially late-acting inbreeding depression in each lineage. Fe was calculated as Fe = (1 − tm)/(1 + tm), where tm is the outcrossing rate calculated from MLTR for each maternal family (Allard, Jain & Workman, 1968; Ritland, 1990).

Inbreeding depression

To estimate inbreeding depression (δ), we collected 150 fruits from different individual plants in each population. These plants did not include the maternal families previously analyzed. From each fruit, we sowed ten seeds on separate pots under greenhouse conditions. Average (±S.E.) environmental variables in the greenhouse (March–September), measured by HOBO sensors (Onset Computer Corporation, Bourne, MA, USA) were: temperature 27.10 ± 0.06 °C; relative humidity 42.93% ± 0.09% and light intensity 3,133.59 ± 18.73 Lux (Camargo, 2009). When seeds germinated, one seedling per fruit was randomly chosen and grown under controlled conditions until flowering. For each population, 100 individuals were randomly chosen to act as pollen receptors (mothers) and 50 individuals as pollen donors (fathers). Two manual pollination treatments were applied to each maternal plant: (1) cross-pollination (o), where two flowers were emasculated before anthesis and hand-pollinated with pollen from one donor randomly chosen from the same population; (2) self-pollination treatment (s), where two flowers of each receptor plant were fertilized with self-pollen. After pollination, flowers in both treatments were bagged individually with a fine nylon mesh. Since many mother plants did not produce the four flowers needed for pollination treatment application, the final sample included mother plants that produced at least one fruit per treatment (CM: N = 77; Map: N = 41). Two fitness components per pollination treatment were evaluated in each population: seed-set mean (i.e., number of seeds/ number of ovules) and seed mass. Seed mass was obtained from a random sample of 30 seeds per fruit using an analytical balance (Adventurer OHAUS). Allocating more resources to seeds can increase quality of seeds, increasing the likelihood of successful seedlings’ establishment and mothers’ fitness (Stanton & Young, 1994; Byers & Waller, 1999). In D. inoxia, as in D. stramonium, under greenhouse experiments conditions, inbreeding depression on seed mass has been detected even when maternal plants have been fertilized with a single pollen donor (Sosenski, 2004; Jiménez-Lobato, 2013). Likely seed mass may influence germination rate and seedling establishment in Datura species.

Cumulative inbreeding depression coefficient (δ) was calculated for each population as: δ=1−w´sw´o,where w´s and w´o are the mean fitness of progenies derived from self- or cross-pollination, respectively. Average fitness of self- and out-cross progenies was calculated as the product of seed-set and seed mass (Schemske & Lande, 1985), and it was used to estimate the primary selfing rate (r) at each maternal lineage. This approach yields an estimate of early-acting inbreeding depression

Statistical analyses

Statistical analyses were implemented in R software version 4.0.2 (R Development Core Team, 2020). To estimate phenotypic variation in herkogamy within each population, we quantified the variance components within and between individuals. Using the nlme package (Pinheiro et al., 2017), we fitted a linear mixed model for each population where maternal family and plants nested within families were considered as random factors.

Statistical differences in primary selfing rates (r), inbreeding in adult cohort (F), inbreeding at equilibrium (Fe) and an estimate of inbreeding depression (Fe–F) between populations were tested by Analysis of Covariance in multcomp package (Hothorn, Bretz & Westfall, 2008). Population was considered as fixed factor and herkogamy as the quantitative variable.

Multilocus outcrossing (tm) and primary selfing rates (r) were strongly negatively correlated (CM: estimate = −1.00, p = 0.000, d.f. = 25; Map: estimate = −0.969, p = 0.000, d.f. = 27), hence we present here only the analyses for primary selfing rates (r). Since r is a proportion, its relationship with herkogamy was analyzed with a beta regression using the betareg package (Cribari-Neto & Zeileis, 2010; Douma & Weedon, 2019). This regression has been proposed for modeling continuous data limited to a specific interval (0, 1) (Ferrari & Cribari-Neto, 2004). We tested for cloglog, logit and log link functions and based on Akaike’s criterion we selected the model that best fit to the data. Estimates of β and ∅ were obtained by maximum likelihood estimation. F, Fe and their differences were associated to herkogamy with a generalized linear model with Gaussian error distribution (Crawly, 2013). It must be noted that Fe is a theoretical prediction based on the selfing rate (r), and these quantities are therefore correlated by construction. Likewise, their correlation with other variables would be very similar. Thus, to avoid redundancy, we present the correlations of selfing rate with other variables.

Results

Variation in herkogamy and mating system parameters within and among populations

At CM plants were, on average, non-herkogamous (mean = −4.72; sd = 5.5 mm; range = 17.48 mm, from −13.83 to +3.65 mm). In this population 20 out of 27 individual plants (74%) had reverse or no herkogamy and seven (26%) exhibited approach herkogamy (Fig. 1A). Herkogamy varied proportionally more among plants at Map than at CM (N = 29) but the average was positive (Mean = 2.57 mm; sd = 11.15 mm; range = 44.8mm, from −20 to +24.8 mm) (Fig. 1B).

Figure 1 Herkogamy value average of individual plants of Datura inoxia: (A) Cañada de Moreno and (B) Mapimí.

Error bars indicate one standard deviation.

The proportion of variance among individual plants in herkogamy was higher in Map (77.32%) than in CM (50.62%), indicating higher intra-individual variation in the latter population (Map: 22.68%; CM: 49.38%).

Analysis of Covariances indicate that primary selfing rates (r), inbreeding at equilibrium (Fe) and inbreeding depression (Fe–F) (Table S1) differed between populations and are affected by herkogamy. Inbreeding in adult cohort (F) was not different between populations and was unaffected by herkogamy (Table S1).

Mating system estimation and inbreeding coefficient (F)

The multilocus outcrossing rate (tm) was, on average, higher in Map than in CM (0.682 vs. 0.294) although high variation in this parameter was detected among lineages in each population (Map, 0.022–1; CM, 0–1). The distribution of tm is skewed toward low values in CM (ca. 60% of mother plants), but towards high values in Map (50% of the families with tm > 0.8) (Figs. 2A and 2B, respectively). Primary selfing rate (r) was two-fold higher in CM than in Map (0.716 vs. 0.353), varying from r = 0 to 1 and from r = 0 to 0.978, respectively.

Figure 2 Histogram and boxplot graphs of outcrossing rate (t) for two populations of Datura inoxia in Mexico: (A) Cañada de Moreno and (B) Mapimí.

Boxplots show quartiles and the median value for each population.

Inbreeding coefficient in the adult cohort (F) was negative in the two populations and highly variable (CM: F = −0.193, range −1 to 0.75; Map: F = −0.085, range −0.553 to 1). Inbreeding coefficient at equilibrium (Fe) was higher in CM than Map (Fe = 0.626 vs. 0.238) and highly variable in both populations (from 0 to 1). The difference between the F of the adult cohort and the Fe of the progeny cohort was much higher in CM than in Map (Fe–F = 0.819 vs. 0.323). The estimated magnitude of cumulative inbreeding depression, measured on greenhouse conditions, was higher in Map than in CM (δ = 0.25 vs. δ = 0.09).

Relationships between herkogamy and mating system parameters

Correlations between primary selfing rate (r), inbreeding coefficient (F), inbreeding coefficient at equilibrium (Fe) and herkogamy were only significant in the Map population (Table 1; Figs. 3 and 4). Primary selfing rate (r) (and hence Fe) were negatively associated with herkogamy only in Map population (Table 1; Figs. 3A and 3B), indicating that individuals with reverse herkogamy or without herkogamy had progenies with higher selfing rate and inbreeding coefficient than plants with approach herkogamy. The best beta regression model between r and herkogamy was fitted with the log link function according to the Akaike’s criterion (Table S2). In addition, the breeding history of each maternal lineage (F) was associated with selfing (r) in the Map population only (Figs. 4A and 4B), but not with herkogamy in either of the two populations (F vs. herkogamy) (Table 1; Figs. 3C and 3D). The Fe – F difference was positively correlated with primary selfing rate (r) only in CM (Table 1; Figs. 4C and 4D). We did not find any indication of a significant correlation between Fe–F and herkogamy (Table 1; Figs. 3E and 3F). As expected, progenies’ inbreeding coefficient (Fe) was positively correlated with primary selfing rates (r) in both populations (results not shown).

Table 1 Correlation between primary selfing rate (r), inbreeding coefficient (adult cohort: F), inbreeding coefficient at equilibrium (progeny cohort: Fe), the difference between Fe and F (Fe–F) and herkogamy in two populations.

Parameter estimates (above diagonal) and their standard error (below diagonal). Significant values are shown in bold type. Statistical models used in each test are described in the Materials and Methods section.

Cañada de Moreno	r	Herkogamy	F	Fe	Fe-F	
r	–	0.023	0.307	0.982***	0.739**	
Herkogamy	0.016	–	−0.001	0.014	0.015	
F	0.234	0.014	–	na	na	
Fe	0.044	0.012	na	–	na	
Fe-F	0.239	0.016	na	na	–	
Mapimí						
r	–	−0.029*	0.572**	0.854***	0.283	
Herkogamy	0.012	–	−0.006	−0.009*	−0.004	
F	0.154	0.005	–	na	na	
Fe	0.042	0.004	na	–	na	
Fe-F	0.158	0.005	na	na	–	
Notes:

* P < 0.05 indicate significant association between variables.

** P ≤ 0.005 indicate significant association between variables.

*** P < 0.0005 indicate significant association between variables.

Figure 3 Relationship between primary selfing rate (r) (A and B), inbreeding coefficient in adult plants (F) (C and D), and inbreeding coefficients at equilibrium (Fe) (progeny cohort; E and F), in relation to herkogamy in plants of Datura inoxia from Cañada Moreno and Mapimí, respectively.

A significant relationship was detected only for r vs. herkogamy in the Mapimí population.

Figure 4 Correlation between inbreeding coefficient (F) (A and B) and the difference between inbreeding coefficient at equilibrium (Fe) and F (Fe–F) (C and D), in relation to primary selﬁng rate (r) in plants of Datura inoxia from Cañada Moreno and Mapimí, respectively.

Model fitting is provided for two significant relationships (B and C), Fitted model is depicted only for the signiﬁcant correlations (B and C).

Discussion

In annual, short-lived, plant species, the evolution of plant mating system in association with flower traits, such as herkogamy, depends on the constancy of natural selection within populations according to the opportunity for cross- and self-fertilization every year (Shirk & Hamrick, 2014). Because herkogamy is a direct modifier of the efficiency of self- and cross-pollination, it is likely to coevolve with the plants’ mating system, and the different within-population lineages would be expected to vary in their history of inbreeding. Here, we evaluated the association of herkogamy, mating system and inbreeding history at a lineage level within two populations of Datura inoxia, an annual/short lived species distributed in arid and semiarid environments in Mexico and southern USA.

We found that herkogamy, selfing rates and inbreeding coefficients, as well as associations among them, varied considerably between and within populations. In line with expectations the Map population, where plants show pronounced approach herkogamy, had higher average outcrossing rate and lower inbreeding coefficient of progenies than individual plants that exhibited absence of and/or reverse herkogamy. These results are in line with the hypothesis of adaptive herkogamy as a mechanism that prevents selfing in populations and avoid inbreeding between individuals (Web & Lloyd 1986; Lloyd, 1992). The relationship between selfing/outcrossing rate and herkogamy has been found in other species of Datura such as D. stramonium (Motten & Antonovics, 1992; Motten & Stone, 2000) and D. wrightii (Elle & Hare, 2002) as well as in other species like Gilia achilleifolia (Takebayashi & Morrell, 2001), Clarkia temblorensis (Holtsford & Ellstrand, 2006), Mimulus ringens (Karron et al., 1997), Nicotiana glauca (Schueller, 2004), Aquilegia canadensis (Herlihy & Eckert, 2007), Gesneria citrina (Chen et al., 2009) and Dalechampia scandens (Opedal et al., 2016).

In the adult cohort of D. inoxia,we did not detect a significant association between the inbreeding coefficient and herkogamy. This result suggests that the inbreeding history of each lineage does not depend on herkogamy alone, and that purging of more inbred individuals and/or biparental inbreeding may also come into play (Charlesworth & Charlesworth, 1987). Reports of low survival rate of inbred individuals at early life, that is, high inbreeding depression, suggests reductions of genetic load in adult plants (Medrano, Herrera & Barret, 2005; Abdelaziz et al., 2014). In fact, in partially selfing populations it has been observed that F in progeny cohorts increase in relation to F of the parental population, and later decrease again in the adult stage (Ritland, 1990; Eguiarte et al., 1993). If the purge of inbred individuals occurs at Mapimí, the relationship between selfing rate in relation to flower herkogamy would be related to the differential contribution of lineages to the next generation, yielding evidence of natural selection on herkogamy (Medrano, Herrera & Barret, 2005).

Although we do not know the causes of the relationship between herkogamy and mating system found in this study, fluctuation of environmental factors, especially pollinator abundance, as it happens at Mapimí, can contribute to the maintenance of variation in herkogamy within populations. Variation in herkogamy is linked with plants’ reproductive assurance when pollinators are scarce, or with high outcrossing rates when abundant (Kalisz, Vogler & Hanley, 2004; Goodwillie, Kalisz & Eckert, 2005; Chen et al., 2009). Since outcrossing rates in Mapimí are related to herkogamy, episodes of differential (or even contrasting) selection on herkogamy among lineages over time may have favored either outcrossing (approach herkogamy) or selfing (no herkogamy and/or reverse herkogamy). Further, phenotypic/genetic variance in herkogamy could facilitate the maintenance of the mixed mating system of D. inoxia in Mapimí. The among individual plants variance component in herkogamy strongly suggests genetic differences between plants, beyond within-individual variance (i.e., heritability > 0). In this sense, our results may provide evidence the herkogamy is a functional trait linked to the reliability of pollinator service in the different populations (Moeller, 2006; Opedal, 2018).

On the other hand, the mating system of the population of Cañada de Moreno is predominantly selfing, and no association between selfing rates, herkogamy and inbreeding coefficients in the adult cohort were detected at this locality. Nevertheless, like the Mapimí population, there is a positive relationship between selfing rate and the Fe–F difference. These results suggest inbreeding depression in selfing individuals followed by genetic purging (Ritland, 1990). The latter explanation is supported by the contrasting average values of inbreeding depression found in the two populations of D. inoxia, being lower in CM than in Map. Theoretical models and experimental results have demonstrated that mutations that cause strong inbreeding depression can be purged from one generation to the next (Willis, 1999; Charlesworth & Willis, 2009), but mutations that cause mild inbreeding depression and are rare can be maintained in populations for multiple generations (Lande, Schemske & Schultz, 1994; Charlesworth & Willis, 2009). The expression of mutations with mild deleterious effects among inbred individuals of D. inoxia can help to explain the differences in inbreeding coefficients between progeny and adult generations and its relationship with selfing rate.

The magnitude of inbreeding depression has been shown to be higher in more stressful than benign environments (i.e., field vs. greenhouse conditions) (Crnokrak & Roff, 1999; Armbruster & Reed, 2005; Cheptou & Donohue, 2011; Fox & Reed, 2011). Both inbred and outcross individuals may perform better under benign environments, and inbreeding depression expresses only as limit and/stressful conditions (Angeloni, Ouborg & Leimu, 2011). Further, it has been predicted that selection against deleterious mutations occurs mainly during the early life history stages and lowers towards later life cycle stages (Angeloni, Ouborg & Leimu, 2011). In this study the assessment of early inbreeding depression under greenhouse conditions could have underestimated the magnitude it can reach in natural, stressful conditions, inhabited by Datura inoxia. Yet, since we did detect early-acting inbreeding depression, results seem to be in agreement with expected trends: CM population, a predominantly selfing population, expresses low value of inbreeding depression than Map, possibly due to a faster purge of deleterious alleles.

Phenotypic variance in herkogamy results from genetic, developmental and environmental factors (Herlihy & Eckert, 2007; Vallejo-Marín & Barrett, 2009; Camargo et al., 2017). However, the evolution of mating system, linked to floral traits like herkogamy, requires additive genetic variance. Additive genetic variance of corolla length and herkogamy has been detected in populations of the annual Datura stramonium (Motten & Stone, 2000; Juárez-Ramírez, 2008; Camargo et al., 2017). Further, a quantitative survey of evolvability of herkogamy measured as mean-scaled additive variance, points out the high evolutionary potential of herkogamy compared to male and female sexual organs or flower size (Opedal et al., 2017). Our results, derived from the partition of phenotypic variance in herkogamy among individual plants and random variation (within individual variation or residual term), indicate a large amount of proportional variance between individuals in each population, but notably more so in Mapimí (77.32%) than in Cañada de Moreno (50.62 %). Thus, there is a strong indication that individual variation in average herkogamy in D. inoxia in Mapimí is genetically based and potentially adaptive (see Jiménez-Lobato & Núñez-Farfán, 2012). High within-individual variation could be adaptive if, on average, high intra-individual variation is linked to higher fitness (Herrera, 2009; Camargo et al., 2017). Otherwise, high intra-individual variation can be maintained in environments that are highly unpredictable in pollinator’s service along time. In the Cañada de Moreno population, the high within-plant variation in herkogamy could limit selection on it, constraining an adaptive response in this population (Falconer & MacKay, 1996; Lynch & Walsh, 1998).

Within-individual variation in plant traits, particularly in flower characters, can be developmental in origin or elicited in response to environmental variability (Herrera, 2009; Camargo et al., 2017). The stability of development or homeostasis has been associated with different levels of heterozygosity, so that heterozygous individuals better buffer environmental variation (Lerner, 1954). However, there is not clear consensus on this hypothesis; some evidence points to the potential effect of inbreeding, with the fixation of deleterious alleles and genetic drift influencing individuals’ level of developmental stability (Clarke, 1993). To what extent inbreeding and deleterious mutations are responsible for intra-individual flower trait variation in the CM population is not known yet, but evidence suggests some developmental variation in flower size and herkogamy, as shown in one highly inbred population of D. stramonium, is linked to environmental variation (Camargo et al., 2017)

Conclusions

Associations between herkogamy, mating system and inbreeding history at a lineage level are expected to occur within populations of self-compatible, hermaphroditic plant species. In D. inoxia, as in other species of Datura, approach herkogamy is associated with higher outcrossing rates and low levels of inbreeding in progeny. The results of this study show that populations of D. inoxia are diverging in mating system characteristics with important genetic implications.

Supplemental Information

Supplemental Information 1 PCR Protocols.

Click here for additional data file.

Supplemental Information 2 ANCOVA to test differences between two populations of Datura inoxia and the effect of herkogamy on: A. Primary selfing rate (r); B. Inbreeding coefficient in the adult cohort (F); C. Inbreeding at equilibrium (Fe) and D.

Click here for additional data file.

Supplemental Information 3 Results of Akaike’s Information Criterion (AIC) to test different link functions for beta regression models between r and herkogamy in Mapimí population of Datura inoxia.

Click here for additional data file.

Supplemental Information 4 Mating system parameters of Datura inoxia families from two Mexican populations.

Herkogamy, outcrossing rate, selfing rate, primary selfing rate, and inbreeding coefficient at equilibrium of families of Datura inoxia from two localities in Mexico.

Click here for additional data file.

We thank the members of the Laboratory of Ecological Genetics and Evolution for field assistance, particularly to Armando López Velázquez. Thanks to Rosalinda Tapia López for lab assistance and to Rafael Torres and Adriana Pérez for greenhouse work assistance. We are indebted to Øystein H. Opedal, Samuel Carleial and one anonymous reviewer for the critical review, and editing, that greatly improved this contribution. We are particularly grateful to Kiko Herrera’s family in the Mapimí Biosphere Reserve and to Tito Galván and family in Cañada de Moreno. This study is part of the Ph. D. thesis of V. Jiménez-Lobato.

Additional Information and Declarations

Competing Interests

Author Contributions

Field Study Permissions

Data Availability

The authors declare that they have no competing interests.

Vania Jiménez-Lobato conceived and designed the experiments, performed the experiments, analyzed the data, prepared figures and/or tables, authored or reviewed drafts of the paper, and approved the final draft.

Juan Núñez-Farfán conceived and designed the experiments, analyzed the data, authored or reviewed drafts of the paper, and approved the final draft.

The following information was supplied relating to field study approvals (i.e., approving body and any reference numbers):

Subsecretaría de Gestión para la Protección Ambiental, Secretaría del Medio ambiente y Recursos Naturales, México approved field collections (SGPA/DGGFS/712/1596/17).

The following information was supplied regarding data availability:

Data is available in the Supplemental Files.

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
