# Peer review of "Mating system of Datura inoxia: association between selfing rates and herkogamy within populations"

_PeerJ, doi:10.7717/peerj.10698_

## Round 0.1 · original submission · Minor Revisions

I leave it to the authors' discretion to evaluate the reviewer's comments and respond as they feel appropriate. For the fastest turnaround, please clearly explain in the rebuttal letter where the manuscript has or hasn't been changed in a point-by-point response to each of the reviewers' comments.

·

Basic reporting

Written text

The manuscript is well-written in general, but a fine tuning in the written English should improve some details in the text. I made some suggestions over the manuscript with track changes. I don’t believe it needs a native speaker or professional editing for the language revision.

Also, please revise typos and missing commas. Example "et al.," and not "et al." Revise terms in italics, such as “i.e.,”. Revise values and statistics regarding empty spaces between values and symbols.


Table
For Table 1, herkogamy should be written in full. There is no information on which side is for which population. Besides, Table 1 is referred in the text when authors discuss on progenies and lineages (ie, basically the first four lines or the other lines, but separately). To me, it is confusing. It would be good if in the Table, not only the population information is cited, but also the progeny or lineage "test" aspect of the comparisons.

Figure in general
Figures represent well the results. However, much improvement is needed. In my opinion:
- there should be a standard way to refer to single panes in the figures (use of letter A, B, etc across figures should be similar).
- Colored bars in grey can be spared for simplicity and for less ink use.
- Make sure to produce graphs with similar font types and sizes. Some axes are represented with values in bold and normal font type, others are in bold and rotated by 90 degrees, for instance.
- P values shown are not equal to 0. Thus, they should be reported as for example P < 0.001 and not P = 0.00.
- Non-significant slopes (lines) should actually not be plotted in my understanding (see Good, P. I., & Hardin, J. W. (2012). Common errors in statistics (and how to avoid them). John Wiley & Sons.). In case authors find it necessary, maybe distinguish them from significant correlation lines making them dashed or in lighter color.
Specific figures

Fig. 1: it seems that the tip of error bars are incomplete (only half) in some cases. Vertical bars are at some points separated by white empty spaces and other times not.

Fig 2: and its caption should be improved. Labels for areas (population) should be centered and reduced in font size. X Axis should better represent the values for the histogram bars (the way it is now is a bit misleading). Besides, the caption should clarify that for each population there are two graphs (boxplot and histogram). At first glance the separate plots seem to be isolated. Color for bars and boxplots could be spared.

Fig. 3 and 4: specify which correlation estimation used in caption (ie. Pearson’s correlation…), consider including the slope/R2 in the significant regression lines. Letters A-H were not mentioned in the caption. In Fig 3, dots may be set as translucent to allow overlaying points to be seen better.

Experimental design

• The submission should clearly define the research question, which must be relevant and meaningful. The knowledge gap being investigated should be identified, and statements should be made as to how the study contributes to filling that gap.

The final paragraph of the Introduction section sounds to me more like a Methods (ie, species and area description), and as such this info should be merged with the info in Methods. Additionally, it would be perhaps meaningful to formulate a better Introduction end, by stating the main gap in the knowledge and the objective of the study.

• Methods should be described with sufficient information to be reproducible by another investigator.

A statistical analyses section should be created and more detailed in my view. It is fragmented between sections, and thus a bit repetitive (with regards to software used). Moreover, I made some comments in the annotated manuscript

Validity of the findings

• Replication experiments are encouraged (provided the rationale for the replication, and how it adds value to the literature, is clearly described); however, we do not allow the ‘pointless’ repetition of well known, widely accepted results.

Authors could discuss on the presence of other populations or related species that might have been added to the story about the evolution of selfing in Datura. This would contribute to the Discussion section in my view.

Additional comments

CM and Map abbreviations are actually not always used across the text.

Comments and suggestions by line
line 20: "Nevertheless, despite the potential advantage of selfing, it may restrict gene flow within and among populations producing high levels of inbreeding between individuals and increasing homozygosity (Ho), and thus the likelihood of inbreeding depression (ẟ)"
In my opinion this long sentence could be reformulated and more clearly be contextualized. For example: Despite the potential transmission/reproductive advantage, selfing as a mating system may as well restrict gene flow within and among populations. Consequently, selfing might increment levels of inbreeding and homozygosity (Ho), and ultimately the likelihood of inbreeding depression (ẟ) in populations.

line 47: "(i.e. no herkogamy)"
Is it the same as nil herkogamy. Nil herkogamy was not defined in the manuscript.

line 230: "To estimate phenotypic variation in herkogamy for each population, we calculated the within- and between-individual variances. Individual plants were considered a random effect. The residual variance is accounted by the intra-plant variation. The significance of individuals was evaluated by a likelihood-ratio ꭓ2 test (LRT) between a model that includes individuals as random effect (function lme) and one that does not (LRT1; function gls) (Zuur et al., 2009). Analyses were conducted with nlme package (Pinheiro et al., 2017) in R software (R Development Core Team, 2008)."
It is not clear what this step is testing and why. At this point, I didnt understand why authors had to compare differences between individuals, but rather herkogamy vs. populations for instance. Since individuals are taken as random effects, how can they be tested for significance rather than being assigned as a fixed-effect predictor? Thus, it needs a better clarification.

line 240: "...glasshouse..."
Maybe describe the conditions of light/humidity/temperature

line 248: "Since r is a proportion with binomial distribution, correlation with herkogamy was calculated by a generalized linear model with a quasi-binomial distribution error (Crawly, 2013)."
I understand that the calculation of correlations using proportional data pose a mathematical problem compared to traditional continuous data correlations. I am not completely convinced, however, about the method used, since the use of quasibinomial distribution assumes a particular variance structure, but this was not mentioned in the methods throughly. The reference on Crawly (2013) doesn't seem to actually explain the use of quasi-binomial distribution on testing correlation. My suggestion would be to (1) better fundament the use of the method (i.e., linear model and distribution) or (2) use another distribution that clearly fits into 0-1 such as beta/binomial (for example, in R one might use the betareg package to run a beta regression).

line 307: "To estimate inbreeding depression (ẟ), we collected 150 fruits from different individual plants in each population"
How many plants and did they include the tagged ones from the measures above? If so, the proportion of those would be interesting to know.

line 381: "This suggests that the inbreeding history at each lineage not only depends on herkogamy; instead, purge of inbred individuals and/or biparental inbreeding may also come into play."
However, they missed to discuss why it was the case by suggesting possible explanations and references that tackle this issue

line 398: "On the other hand, the mating system of the population of Cañada de Moreno is predominantly selfing, and no association between selfing rates, herkogamy and inbreeding coefficients in the adult cohort were detected."
However in Table 1, we see that

line 432: "pollination likelihood by animals" what does this mean. Better formulation

·

Basic reporting

The authors report on a study of two populations of Datura inoxia in Mexico, aimed at quantifying variation in floral traits, mating patterns, and inbreeding among individual lineages within each population. These associations have been well studied in a range of systems, but as the authors point out, relatively few studies have assessed these associations within (rather than among) populations.

The authors have assembled a rather large dataset on flower morphology, outcrossing rates and inbreeding coefficients derived from microsatellite markers, and have also quantified inbreeding depression experimentally in the greenhouse. I found the data collection to be adequate, and the dataset to be appropriate for quantifying the focal variables.

Most of my comments relate to the presentation of the manuscript, which I found less than ideal. Most of this is language issues, and could thus be fixed by revising the text.

Experimental design

The data collection protocols appear adequate to me, although the presentation could be improved in some places as outlined in my line-by-line comments.

In the analyses, I suggest considering testing for differences between populations in ANCOVA-type analyses (y ~ herkogamy*population). In the results can then focus more on the contrast between the two populations, than on the statistical significance of patterns within each population.

Validity of the findings

I agree with most of the authors conclusions based on the study, but have suggested some areas where further elaboration could increase the impact of the paper.

The relationship between individual-level herkogamy and mating patters differs a bit from the expectations for among-population relationships. Among populations we except herkogamy-outcrossing relationships to have evolved in response to variation in, e.g., the reliability of pollination. Within populations, a correlation between herkogamy and outcrossing rate could relate to processes such as stigma clogging and related mechanisms of sexual interference, as originally discussed by Lloyd and Webb 1986. I suggest expanding a little on the specific mechanisms in play at the within vs. among-population level. A recent review of these ideas is given in Opedal 2018 IJPS.

Unpredictable pollination service is casually discussed as a potential source of (fluctuating) selection on mating systems, but it is not so clear whether this relates to daily fluctuations vs. variation within and among seasons. The link from fluctuating selection (at the population level) to maintenance of individual lineages with contrasting mating histories is not that clear to me. Selection for reduced herkogamy as a mechanism of reproductive assurance, sure, but why would not the population reach an ‘optimum’ herkogamy, with some phenotypic variation around it (due to plasticity etc.). At least these issues could be discussed more explicitly.

The data allows estimating (early-acting) inbreeding depression in the greenhouse, and also (potentially late-acting) inbreeding depression in the field (the comparison of F to Fe, as outlined by Ritland 1990). This could be made a more explicit part of the presentation, as it is currently unclear how often these two approaches give similar results.

Additional comments

Some suggestions for clarification of individual points and sentences:

Abstract
I found it a little confusing to read about ‘genetic variance in herkogamy’ in the Abstract, as this does not seem to be addressed

Introduction
Line 2: Plant mating system affect the demographic dynamic and genetic structure of populations, and thus their adaptive potentials.
Line 4: Mating systems often vary widely among populations of self-compatible species.
Line 9: Maybe start directly from: ‘The transition from…’
Line 14: Two main mechanisms are thought to favour the evolution of selfing
Line 15: ‘genomes’ -> gametes
Line 41: See also Opedal 2018 IJPS for a review of the expected functional relationships between herkogamy and mating patterns, and a review of the literature.
Line 43: Herkogamy is a complex trait determined by…
Line 53: There are a few more studies, see the review in Opedal 2018 IJPS
Line 57: Maybe change heterogeneous to stressful?
Line 74: What suggests that the variation in adaptive? Association with different pollination environments?
Line 77-78: Or selection on herkogamy which affects the mating system and generates an inbreeding history? Not sure correlated selection needs to be invoked here.

Methods
Line 81: ‘Study species and sites’?
Line 85: September, similar to other Datura species
Line 88: stigma receptivity
Line 88: from a few up to
Line 90: Flower traits … vary widely within and among populations
Line 92: forage for nectar
Line 97: Maybe ‘herkogamy, selfing rate, and inbreeding coefficients’ is more logical?
Line 99: ‘e.gr’ - > including
Line 101-102: ‘indicate substantial within-population variation’
Line 110-102: Please clarify the focus here – daily fluctuations or seasonal variation?
Line 116: Measured as, or calculated as?
Line 117: Mention how approach/reverse herkogamy translates into positive/negative values as recorded in this study.
Line 121: I don’t think there is any need for this statistical hypothesis test. Simply quantifying the variance components should suffice (possibly with confidence intervals).
Lines 132-136: If 20 seedlings per maternal family was analysed, no need to point out that 25 were initially collected.
Line 142: shown -> described. (And thank you for that!)
Line 152-160: I like the analysis of primary selfing rates!
Line 161: Why is biparental inbreeding not a part of the mating system?
Line 169-170: ‘larger differences’?
Line 172: Consider citing Allard et al. 1968 here, where the equivalent formulation s/(2-s) appears.
Line 174 and throughout: multilocus outcrossing rate
Line 177: Is the herkogamy-outcrossing relationship ‘just a correlation’, or a causal relationship?
Line 178: error distribution
Line 184: on -> in. Please specify greenhouse growth conditions.
Line 186: reproduction -> flowering
Line 188: maternal plant
Line 189: hand-fertilized -> hand-pollinated
Line 195: fitness components

Results
Line 205: Variation in herkogamy within and among populations
Line 206: At Cañada de Moreno, plants were on average approach herkogamous, but with substantial variation (mean = m range =).
Line 208 and throughout: Consider replacing ‘showed approach herkogamy’ with something like ‘were approach herkogamous’ or ‘exhibited approach herkogamy’. Reads better at least to me
Line 212: Herkogamy varied proportionally more among plants at Mapimí than at Cañada…
Lines 214-216: I suggest dropping this hypothesis test. The variance components speak for themselves.
Line 221: to -> towards
Line 229-230: Make the distinction between the greenhouse and population-genetic estimate of inbreeding depression clearer.
Line 232: Relationships between herkogamy and mating-system parameters
Line 233 Maybe focus more on the difference between populations than on the statistical significance within each. Consider even testing for differences in an ANCOVA-type analysis (y ~ herkogamy*population)
Line 240-241: So the take-home here is that herkogamy is unrelated to the breeding history of the population? This would be highly surprising at least to me

Discussion
The discussion overall flows better than e.g. the introduction.

Line 249-250: Sounds a little strange that it is the ‘association between the traits and the mating strategies’ that is favored. Perhaps rephrase to reflect that certain traits (like herkogamy) is likely to coevolve with the mating system, because it is a direct modifier of the efficiency of self and cross-pollination?
Line276: Maybe explain -> facilitate?
Line 297: Very inaccurate. Rewrite to reflect the relationships between additive variance, evolvability (mean-scaled additive variance) and heritability (variance-scaled additive variance). What Opedal et al. 2017 argue is that herkogamy is highly evolvable on a proportional scale, i.e. it’s easy to change herkogamy by e.g. 10%, compared to changing e.g. flower size by 10%. This is in part due to relatively weak constraints arising form the covariation between pistil length and stamen length in many systems. Note also the high evolvability estimate for D. stramonium presented in Opedal et al. 2017, could be cited in support of the present findings of substantial among-family variance.
Line 301: Point out that these percentages represent proportional variation (because the two populations differ in phenotypic variance, the percentages do not map directly to variation in units of mm).

Table 1: Why not present this as a correlation matrix with standard errors/support values on one of the diagonals?

Figure 1: Eliminate the x-axis labels, as mostly confusing?

Figure 2: Do the box plots show means or medians (normally medians)?

Figure 3: Why are linear trend lines shown if analyses were done with binomial GLMs?

Figure 4: The hypothesis test seems strange for the deterministic relationships shown in C and D.

Reviewer 3 ·

Basic reporting

no comments

Experimental design

The experimental design is generally standard and consistent with previous research, except for some potential flaws that may require concerns.

1. Estimation of the inbreeding depression may be biased. Inbreeding depression is estimated based on the artificial pollination treatment. It is quite likely that an individual with a higher selfing rate will also have a lower number of deleterious mutations in the genome, which means a higher fitness. Given there is a high variation in selfing rate, this correlation may be strong.

However, this correlation between selfing rate and fitness is not reflected in the artificial pollination treatment. Therefore, inbreeding depression estimated using the artificial pollination experiment will be different from the inbreeding depression when pollination happens naturally. This inconsistency affects the calculation of the primary selfing rate, as the seeds used for estimating selfing rate are collected from the natural population.

2. The accumulative inbreeding depression (δ) used to calculate the primary selfing rate (r) is problematic. In the equation r=s/(1-δ+δs), δ accounts for the selfed offspring died from the stage of fertilization to the stage of the measure of the selfing rate s in the offspring seedlings.

However, it is said the cumulative inbreeding depression has two components: mean seed set and seed mass. Why can seed mass be used as a component of fitness? Is there any evidence in Datura inoxia showing a linear relationship between seed mass and the probability for a seed to germinate to a seedling? If not, then seed mass will not play a role in reducing the selfed offspring at the stage of the measure of the selfing rate, and should not be included into the cumulative inbreeding depression.

Validity of the findings

Generally the resutls presented are good, but there is one suggestion:

1. It may be unnecessary to show a correlation between herkogamy and the equilibrium inbreeding coefficient F_e in Fig. 3. F_e=s/(2-s) is a theoretical prediction based on the selfing rate s. Therefore, given the correlation between the primary selfing rate r and herkogamy has been already shown, it seems redundant to show a correlation for F_e.
The same logic also applies to Fig. 4, where showing a correlation between F_e and the primary selfing rate r is of no value, as it is almost guaranteed by the calculation of F_e. Also, based on the equation F_e=s/(2-s) and s=(1-δ)r/(1-δr), it is clear F_e will not have a linear relationship with r.

Additional comments

Generally, the paper is good and a standard investigation on the relation between phenotype, selfing rate, inbreeding coeffcient and inbreeding depression. It may be interesting to have an explanation why inbreeding coeffcient is negative.

Annotated reviews are not available for download in order to protect the identity of reviewers who chose to remain anonymous.

---

## Round 0.2 · Minor Revisions

I appreciate the care and thoughtfulness with which the reviewers' concerns have been addressed. I am returning this solely to give the authors the opportunity to address the handful of minor suggestions for improvement in the comments from Reviewer 1 and in the annotated manuscript from Reviewer 2. I don't anticipate that the next revision would need to go to external reviewers again.

·

Basic reporting

I believe the authors adequately accepted/answered the issues raised in my previous feedback. After the revisions accounting for the comments and suggestions of the reviewers (including myself), I consider that the manuscript is adequate for publication. The written text and illustrations are improved and to my opinion ready for publication.

Two minor details, however, I would still raise for fine tuning (regarding text format only):

1) line 127: "error SD". I believe authors refer to standard deviation error. This notation should be the same as for the other notations across the text, which were noted as "sd" (lower case). Figure 1 caption also uses SD (uppercase)

2) line 203: "Statistical analysis" is a subsection of "Material and Methods", and therefore not bold, but rather in italics.

Finally, I would make one final suggestion, but I leave it to the authors to decide since it is not a fundamental point. Since figures comparing the two populations are more clean now, maybe using a nice colour code could help distinguish the two populations more easily.

Experimental design

no comments

Validity of the findings

no comments

Additional comments

no comments

·

Basic reporting

See below

Experimental design

See below

Validity of the findings

See below

Additional comments

The authors have done a good job revising the manuscript, which is clearly improved from the previous version. I have made only minor comments and suggestions on the attached file. The text could benefit from some final polishing. I congratulate the authors on an interesting study!

Reviewer 3 ·

Basic reporting

no further comment

Experimental design

no further comment

Validity of the findings

no further comment

---

## Round 0.3 · accepted · Accept

This is a rigorously conducted and novel observational study on intra-specific variation in herkogamy and its evolutionary causes and consequences in a natural plant population.